# Anti-Predation Responses to Conspecific versus Heterospecific Alarm Calls by the Nestlings of Two Sympatric Birds

**DOI:** 10.3390/ani12162156

**Published:** 2022-08-22

**Authors:** Yuxin Jiang, Jingru Han, Canchao Yang

**Affiliations:** Ministry of Education Key Laboratory for Ecology of Tropical Islands, College of Life Sciences, Hainan Normal University, Haikou 571158, China

**Keywords:** call recognition, eavesdropping, mobbing alarm calls, parent–offspring communication

## Abstract

**Simple Summary:**

Birds can adjust their behavior by reacting to sound cues. Parents can adjust the feeding rate of nestlings by their response to begging calls. Acoustics play an important role in parent–offspring communication. When adults perceive predators, they can make mobbing alarm calls to warn other individuals. Chicks can also respond to the alarm calls of parents and thus reduce their predation risk. We tested whether the chicks of two species could hear, and respond to, conspecific and heterospecific acoustic alarm signals. Chicks were found to eavesdrop on conspecific and heterospecific mobbing alarm calls. Acoustical similarities between the alarm calls may explain why chicks can recognize heterospecific calls.

**Abstract:**

Predation is generally the main cause of bird mortality. Birds can use acoustic signals to increase their predation survival. Bird response to mobbing alarm calls is a form of anti-predation behavior. We used a playback technique and acoustic analysis to study the function of mobbing alarm calls in the parent–offspring communication of two sympatric birds, the vinous throated parrotbill (*Sinosuthora webbianus*) and oriental reed warbler (*Acrocephalus orientalis*). The chicks of these two species responded to conspecific and heterospecific mobbing alarm calls by suppressing their begging behavior. The mobbing alarm calls in these two species were similar. Mobbing alarm calls play an important role in parent–offspring communication, and chicks can eavesdrop on heterospecific alarm calls to increase their own survival. Eavesdropping behavior and the similarity of alarm call acoustics suggest that the evolution of alarm calls is conservative and favors sympatric birds that have coevolved to use the same calls to reduce predation risk.

## 1. Introduction

Predation is the leading cause of death in many species, and many birds and mammals use alarm calls to warn conspecifics of danger [1,2]. Individuals can communicate different hazardous conditions by adjusting the acoustic parameters of their alarm calls [3,4]. For example, meerkats (*Suricata suricatta*) can change the acoustic parameters of alarm calls depending on the species and behavior of the predator [5]. Animals that cannot vocalize may also use sound to reduce the risk of predation. The lizard (*Liolaemus lemniscatus*) can reduce its activity in response to sound [6]. Alarm calls and types vary greatly among species and include the following: (1) Flee alarm calls, which are given in response to urgent danger from predators, leading other individuals to become cryptic or to flee; (2) mobbing alarm calls, which are given to predators not posing danger directly, leading other individuals to approach and monitor or harass the predator; and (3) distress calls, which are given when an individual is attacked or captured, possibly startling the predators or recruiting others to help the caller [7]. Predation risk may select for the evolution of complex alarm systems [1]. Many birds can communicate the type, size, and distance of a predator through alarm calls [8,9,10,11]. Alarm calls convey information about predators, which can enhance the visual perception of predators through signal receivers. This indicates that alarm calls can induce objective images about predators in the brain of signal receivers, similar to nouns in human language [12]. Alarm calls can encode other types of danger information. For example, the yellow warbler (*Setophaga petechia*) can produce a “seet” warning of nest parasitic risks from the brown-headed cowbird *(Molothrus ater*). Sympatric red-winged blackbirds (*Agelaius phoeniceus*) can eavesdrop on the anti-brood parasite alarm calls of yellow warblers to reduce their own risk of parasitism [13,14]

The information in alarm calls is understood not only by conspecifics but also by other species [7,15]. For example, grey squirrels (*Sciurus carlinensis*) can eavesdrop on the alarm calls of songbirds to reduce their risk of predation [16]. The eavesdropping behavior of heterospecific alarm calls has been found among birds or mammals, between birds and mammals, and between reptiles and birds [17]. Evolutionary forces have selected similar acoustic parameters for the alarm calls among species, and this similarity allows species to understand each other [18]. Mammals and reptiles are more likely to eavesdrop on birds than other animals. This may be because birds have a wide field of vision and high-decibel alarm calls, so their information is reliable and the sound signals are easy to obtain [7].

Accessing social information from individuals of other species may help animals adapt to changing environments [19,20]. For many species, the use of threat information encoded by other animals against predators reduces the time and energy that individuals allocate to alert activities so that they can devote more time and energy to foraging and other activities [16,21]. The expression of IGE (immediate early gene) in the auditory region of signal recipients increases with the increase in the threat level encoded by the conspecific or heterospecific species [22]. This indicates that the level of danger corresponds to the level of neural activity in the auditory region. The same neural activity in the eavesdropper can be induced by the same and different alarm calls. Eavesdropping on other species can provide more information than eavesdropping on the same species because differences in the perceptual abilities and habitat space use of species extend the limited perceptual space covered by a single species [23]. However, heterospecific alarm calls are not always reliable. Therefore, some birds will identify the source of relevant information and encode the predator information that can be directly obtained, and not encode the heterospecific alarm calls. For example, red-breasted nuthatches (*Sitta canadensis*) vary their mobbing calls to reflect the predator threat, and when nuthatches obtain indirect information, they produce calls with intermediate acoustic features. This indicates that although heterospecific eavesdropping is effective, it still needs to be balanced by the reliability of the information [15,19].

The alarm calls of adult birds can alert other adults and also chicks. For example, chicks of the noisy miner (*Manorina melanocephala*) can recognize conspecific land and air alarm calls and adjust their begging intensity according to the different alarm calls [24]. In addition to adjusting the intensity of begging, nestlings have other behaviors that are responses to alarm calls [25]. Chicks of the great tit (*Parus major*) curl up in their nests when hearing “chicka” alarm calls from their parents and leave the nest after hearing “jar” alarm calls related to snake predators. In addition to responding to conspecific alarm calls, nestlings can also respond to heterospecific alarm calls [26]. Parent–offspring communication is important for altricial birds because these nestlings cannot escape predators. Nestlings are highly dependent on their parents for care and cannot recognize predators during their development [27]. In addition, nestlings live in environments where they can eavesdrop on heterospecific alarm calls, further reducing their risk of predation [24]. Studies on nestling responses to alarm calls have focused on conspecific communication, and further research is needed on how nestlings respond to heterospecific alarm calls. Herein, we studied parent–offspring communication and eavesdropping via mobbing alarm calls in the vinous-throated parrotbill (VP) *Sinosuthora webbianus* and oriental reed warbler (ORW) *Acrocephalus orientalis*. These two species have large populations and sympatric breeding habitats, and also share the same predators. We recorded the adult mobbing alarm calls of these two species, analyzed their differences, and determined playback responses to conspecific and heterospecific chicks. The goal of the study was to test whether VP/ORW offspring would be able to recognize and respond to the conspecific/heterospecific mobbing alarm calls of adults.

## 2. Materials and Methods

### 2.1. Study Area and Subjects

The study was carried out in Yongnianwa National Natural Park, Yongnian District, Hebei Province (36°40′60′′–36°41′06′′ N, 114°41′15′′–114°45′00′′ E). The fieldwork was conducted in April to August 2021. The predators of VP and ORW include the Siberian weasel (*Mustela sibirica*), the Erythema snake (*Dinodon rufozonatum*), and the Brown rat (*Rattus norvegicus*) [28]. There is a large population of VP and ORW breeding sympatrically in the wetland. Their nests were distributed in a mosaic pattern within the habitat, with similar nest structures and nest sites in the wetland [29,30].

### 2.2. Production of Playback Sounds

By monitoring the nests of AP and ORW, we recorded the mobbing alarm calls of parents when the chicks were 3 days old. A portable recorder (Sony PCM-A10, Tokyo, Japan) was used to record the mobbing alarm calls that parents uttered to the researcher for 1–2 min (n = 12 for VP, n = 11 for ORW) at a distance of 1 m away from the nest to obtain sufficient sentence samples for analysis. The sampling frequency was set to 44.1 kHz and the sampling accuracy was set to 16 bits (file format: wav). To obtain good-quality recordings for analysis, the distance between the researcher and the bird was less than 5 m. We analyzed 5 acoustic parameters of basic sound (BS) using Raven Pro version 1.4 in each phrase of the distress calls: (1) Duration, (2) lowest frequency, (3) highest frequency, (4) delta frequency, and (5) peak frequency. Based on the data distribution, we conducted Mann-Whitney *U* tests or Student’s *t*-tests for comparisons (Table 1).

To avoid pseudo-replication in playback experiments, we randomly chose three adults from each species and combined their mobbing alarm calls as playback sounds. To obtain high-quality mobbing alarm calls, we used Raven Pro version 1.4 software (Cornell Lab of Ornithology, Cornell University, Ithaca, NY, USA) to chip the sound that we recorded, removed noise below 0.2 kHz, and randomly combined the recordings to form 30 s mobbing alarm calls. We did not change the syllable type or call rate in the segment of mobbing alarm calls. We recorded the background noise from the wetland and randomly chose and chipped the background noise to make three 0.3 s sound fragments, and we randomly combined these fragments to form the 30 s background noise (Appendix A).

We conducted the playback to chicks (clarified in the section below). For the playback experiment, the playback stimuli consisted of three sets of independent sound: 30 s of either VP mobbing alarm calls, ORW mobbing alarm calls, or background noise. Each set of sounds was played at the same distance of 1 m and the same sound pressure level (SPL) by using a sound meter (Smart Sensor, AR824, Dongguan, China). The amplitude was close to the natural level when we recorded the mobbing alarm calls (VP: 54.43 ± 0.64 dB (mean ± standard error); ORW: 74.2 ± 0.65 dB (mean ± standard error); background noise: 48.96 ± 1.26 dB (mean ± standard error)).

### 2.3. Playback Experiments

We chose five-day-old chicks for the playback experiment, and at this stage, they start to produce obvious begging calls. The eyes of five-day-old chicks are barely open, so they would not be alerted by human disturbance. We randomly removed one chick from each nest (n = 17 for VP and n = 18 for ORW) to the indoor lab of the study area to avoid interference such as reactions from nestmates and parents. The chicks were weighed on an electronic scale (Yuedi Electronic scale, Shenzhen, China), placed into an empty nest that was collected during the previous year, and then left alone for 40 min before initiation of the playback experiment. A digital video camera (Sony PCM-A10, Tokyo, Japan) was mounted nearby to record the begging behaviors of chicks, and a Bluetooth speaker (ShiDu P3, Shenzhen, China) was placed at a distance of 1 m from the chicks for playback of the mobbing alarm calls. During the experiment, an observer (Y.J.) simulated parent visitation by lightly touching the edge of the nest (one touch/3 s) to stimulate the begging behavior of chicks. The playback experiment was composed of 30 s of behavior and acoustic recording without playback (natural begging) following the playback stimuli (i.e., playing 30 s of VP mobbing alarm calls, ORW mobbing alarm calls, and background noise in random order). Therefore, the playback experiment included 4 trials (30 s for each trial), and the interval between trials was 5 min for return to the base conditions. The purpose is to compare the begging behavior between the natural state and playback conditions, and the purpose of playing back background noise is to eliminate noise interference factors in the playback sound. We conducted playback with the video and sound recording to quantify 3 aspects of begging behavior: (1) The number of beak-openings, (2) the number of begging calls, and (3) the begging duration (time of beak opening) during each 30 s of observation [28]. Each chick was returned to its own nest after the 1 h experiment. All chicks were accepted by their own parents and were given food by their parents as normal.

### 2.4. Statistical Analysis

We tested the carryover effect produced by the order of presentation due to the recovery time between stimuli. The order of presentation was used as an independent variable in all trials using generalized linear mixed-effect models (GLMMs). The results showed that there was no carryover effect of order presentation in all trials (*p* > 0.05). For the playback-to-chicks experiment, GLMMs with Poisson distribution were used in each dependent variable, in which the response variable was the beak opening frequency, the number of begging calls, or begging duration, while the playback treatment had 4 levels (natural begging, VP and ORW mobbing alarm calls, and background noise). The individual identity was included as a random-effect factor, and body weight was included as a covariate. Chi-square tests were used to obtain *p*-values by comparing the fitted models to null models of random effect. A pairwise comparison was performed with the R package *Agricolae* using the Bonferroni post hoc test. We also used marginal pseudo-R^2^ to evaluate the effect size of fixed effect factors via the R package *MuMIN* [31]. All GLMMs were evaluated using the *lme4* package within R version 4.0.5 (Ross Ihaka; Robert Gentleman, The University of Auckland, New Zealand).

## 3. Results

The begging duration of both VP and ORW chicks differed among the playback stimuli (VP: χ^2^ = 116.89, *p* < 0.001; marginal pseudo-R^2^ = 0.636; ORW: χ^2^ = 163.62, *p* < 0.001; marginal pseudo-R^2^ = 0.52, GLMMs, Figure 1). Both the conspecific and heterospecific mobbing alarm calls reduced the begging duration in the chicks of the two species compared with natural begging and background noise control, but no difference was found in the begging duration between the reaction to conspecific and heterospecific mobbing alarm calls for both species (Figure 1 and Table 2). The number of begging calls also differed among the playback stimuli (VP: χ^2^ = 109.23, *p* < 0.001; marginal pseudo-R^2^ = 0.45; ORW: χ^2^ = 127.27, *p* < 0.001; marginal pseudo-R^2^ = 0.36, GLMMs, Figure 2). Both species of chicks reduced the number of begging calls after receiving conspecific/heterospecific mobbing alarm calls compared with natural begging and background noise, but there was no difference in the number of begging calls to conspecific and heterospecific mobbing alarm calls in either species (Figure 2). For the VP, there was no difference in the number of beak-openings among the playback stimuli (χ^2^ = 0.5, *p* = 0.92; marginal pseudo-R^2^ = 0.007, GLMMs; Figure 3. However, for the ORW chicks, the conspecific mobbing alarm calls reduced the number of beak-openings compared with the heterospecific mobbing alarm calls, natural begging, and background noise (χ^2^ = 7.64, *p* = 0.05; marginal pseudo-R^2^ = 0.1, GLMMs; Figure 3). Body weight did not affect the three response variables of VP (*p* > 0.05, GLMMs), and also did not affect the beak opening frequency of ORW (*p* > 0.05, GLMMs). However, body weight had a significant effect on the begging duration (*p* = 0.002, GLMMs) and the number of begging calls (*p* = 0.009, GLMMs).

## 4. Discussion

The results showed that conspecific/heterospecific mobbing alarm calls suppressed the number and duration of begging calls in VP and ORW chicks. This suggested that chicks of both species can hear conspecific and heterospecific alarm calls and respond appropriately to reduce predation risk. Chicks often inhabit complex environments, so they may further reduce the risk of predation by eavesdropping on the alarm calls of other species. Eavesdropping of conspecific/heterospecific mobbing alarm calls may be an innate mechanism or a learned response, providing important opportunities for obtaining additional information about predators [32]. Compared with the natural begging and playback stimuli, chicks of VP did not suppress their beak opening frequency when they heard conspecific or heterospecific mobbing alarm calls. However, ORW chicks suppressed their beak opening frequency when receiving conspecific mobbing alarm calls. This indicated that ORW chicks may discriminate conspecific mobbing alarm calls from other playback stimuli. Chicks of ORW are larger than chicks of VP, and begging visual signals are more obvious to predators and face stronger natural selection. The chicks of VP maintained the short visual signal intensity of begging and reduced the signal level of begging calls. This may indicate that the chicks do not completely stop begging after receiving danger information, but balance begging with predation risk by adjusting the intensity of their begging behavior. Alternately, unlike acoustic signals such as the number of begging calls and persistent visual signals such as begging duration, the beak opening frequency could be a short visual signal produced by chicks without sound. The short visual signal may be safer than using an acoustic signal or long visual begging, because the transmission of a visual signal would be blocked by vegetation, becoming less effective but more cryptic [33]. Similar to the function of distress calls, mobbing alarm calls inhibited their begging behavior to reduce the risk of predation [28].

In altricial birds, chicks cannot actively flee the nest when they encounter predators and are highly dependent on parental care. Therefore, effective parent–offspring acoustic communication is an important way for altricial birds to avoid predation. A previous study found that chicks responded to two different alarm calls produced by conspecifics but did not respond to the alarm calls of a different species in a sympatric area [24]. This is not consistent with the results of our study. Chicks of VP and ORW responded to heterospecific mobbing alarm calls by suppressing begging behavior. This may be related to the fact that there was no significant difference between the peak frequency and lowest frequency of basic sound due to the conservative evolution of the alarm calls. It also suggests that the chicks may show inhibitory begging behavior, because of the similarity between acoustic signals, rather than learning behavior.

## 5. Conclusions

The results of a playback experiment showed that the chicks of VP and ORW can hear both conspecific and heterospecific mobbing alarm calls and respond by suppressing their begging behavior. This response reduces the risk of predation and implies that mobbing alarm calls play an important role in parent–offspring communication. It can increase reproductive success and enhance bird fitness.

## Figures and Tables

**Figure 1 animals-12-02156-f001:**
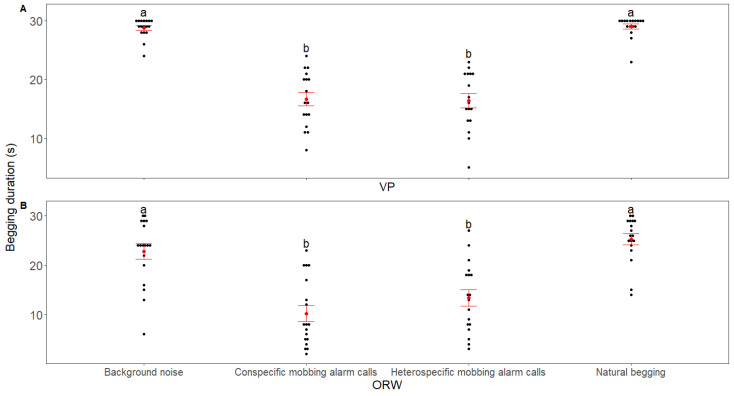
Comparison of the begging duration from nests as a response toward playback stimuli between (**A**) VP and (**B**) ORW in the playback experiment. The red points and whiskers represent the mean and standard errors of the observed data, respectively. The black points represent the raw data, while treatments with the same/different letters indicate the nonsignificant (*p* ≥ 0.05)/significant (*p* < 0.05) differences in responses, respectively.

**Figure 2 animals-12-02156-f002:**
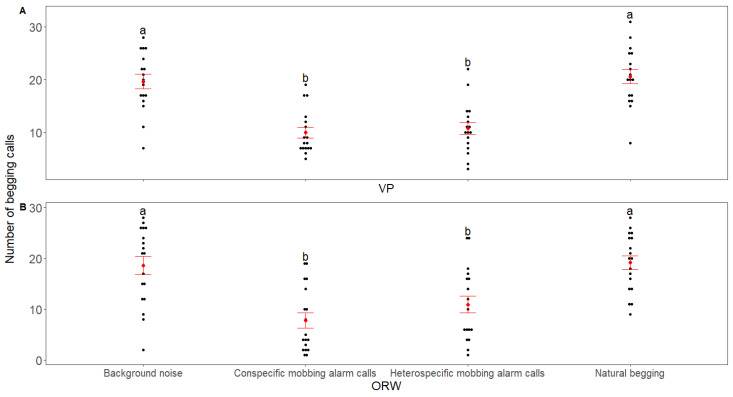
Comparison of the number of begging calls from nests as a response to playback stimuli between (**A**) VP and (**B**) ORW in the playback experiment. The red points and whiskers represent the mean and standard errors of the observed data, respectively. The black points represent the raw data, while treatments with the same/different letters indicate the nonsignificant (*p* ≥ 0.05)/significant (*p* < 0.05) differences in responses, respectively.

**Figure 3 animals-12-02156-f003:**
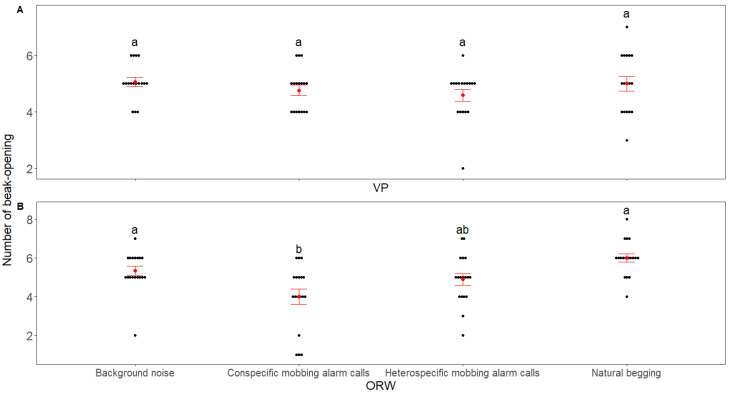
Comparison of the number of beak-openings from nests as a response toward playback stimuli between (**A**) VP and (**B**) ORW in the playback experiment. The red points and whiskers represent the mean and standard errors of the observed data, respectively. The black points represent the raw data, while treatments with the same/different letters indicate the nonsignificant (*p* ≥ 0.05)/significant (*p* < 0.05) differences in responses, respectively.

**Table 1 animals-12-02156-t001:** Comparison of five acoustic parameters of basic sound of adults’ mobbing alarm calls between the vinous-throated parrotbill (VP) and the oriental reed warbler (ORW).

Parameter of Phrase	VP (n = 12)	ORW (n = 11)	w	*p*
Highest frequency (Hz)	6000.17 ± 332.89	6601.91 ± 178.96	32	0.04
Delta (Hz)	3945.6 ± 306.27	4535.47 ± 190.47	24	0.008
			**t**	** *p* **
Duration (s)	0.08 ± 0.004	0.15 ± 0.023	−2.91	0.02
Peak frequency (Hz)	4772.52 ± 142.69	4588.59 ± 111.005	1.02	0.32
Lowest frequency (Hz)	2154.57 ± 109.06	1966.45 ± 110.92	1.21	0.24

Comparisons were analyzed by the Mann-Whitney U test or Student’s *t* test.

**Table 2 animals-12-02156-t002:** Results of generalized linear mixed models for the responses in the experiment of playing back mobbing alarm calls to the chicks of the vinous-throated parrotbill (VP) and the oriental reed warbler (ORW).

Response Variable	VP				ORW		
	SE	Z	*p*		SE	Z	*p*
Begging duration, marginal pseudo-R^2^ = 0.636 (VP) and 0.52 (ORW)							
Intercept	0.05	73.03	<0.001	Intercept	0.08	35.59	<0.001
Conspecific mobbing alarm calls	0.07	−7.67	**<0.001**	Conspecific mobbing alarm calls	0.09	−10.4	**<0.001**
Background noise	0.06	−0.16	0.83	Background noise	0.08	−7.99	0.13
Heterospecific mobbing alarm calls	0.07	−7.47	**<0.001**	Heterospecific mobbing alarm calls	0.08	−1.5	**<0.001**
Number of begging calls, marginal pseudo-R^2^ = 0.45 (VP) and 0.36 (ORW)							
Intercept	0.08	35.29	<0.001	Intercept	0.12	24.69	<0.001
Conspecific mobbing alarm calls	0.09	−7.13	**<0.001**	Conspecific mobbing alarm calls	0.1	−9.03	**<0.001**
Background noise	0.08	−0.61	0.54	Background noise	0.08	−6.3	0.67
Heterospecific mobbing alarm calls	0.09	−7.79	**<0.001**	Heterospecific mobbing alarm calls	0.09	−0.42	**<0.001**
Beak opening frequency, marginal pseudo-R^2^ = 0.01 (VP) and 0.2 (ORW)							
Intercept	0.11	14.84	<0.001	Intercept	0.01	18.62	<0.001
Conspecific mobbing alarm calls	0.16	−0.55	0.58	Conspecific mobbing alarm calls	0.15	−2.67	**0.01**
Background noise	0.15	−0.08	0.93	Background noise	0.14	−1.43	0.15
Heterospecific mobbing alarm calls	0.16	−0.31	0.76	Heterospecific mobbing alarm calls	0.14	−0.84	0.4

Natural begging calls were the baseline for response; *p* values (for model variables) < 0.05 are highlighted in bold. Pseudo-R^2^ refers to the effect size.

## Data Availability

The data presented in this study are available on request from the corresponding author.

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
