# Peer review of "Anti-Predation Responses to Conspecific versus Heterospecific Alarm Calls by the Nestlings of Two Sympatric Birds"

_animals, 2022, doi:10.3390/ani12162156_

Round 1
Reviewer 1 Report
I enjoy reading ms. My main concern is that methods lack clarity that allow repeat the experiment.
Here my minor comments
Line 108-109: in how many nest you recorded the alarm calls
Line 109: Do you use the internal microphone? was it a mono or stereo recording? File format?
Table 1: n is the number of nests or the number of calls analyzed?
Line 123-124: this sentence is not clear. You mixed in a single playback stimulus call from 3 adults? If this is the case, it is a very uncommon stimulus type, because you are producing stimulus from 3 different individuals, which is not common in nature. So, you need to justify this uncommon stimulus design.
Line 127: which filter type you used
Line 135: which is the SPL value? You need to report the value in dB.
Lines 137-138: 54.43 and 74.2 are? Units
Line 154: a single stimulus has the three stimuli in sequence? Explain this very well
Lines 154-156: these sentences need to be explained in more detail because it is not clear what you did.
Author Response
Reviewer 1:
I enjoy reading ms. My main concern is that methods lack clarity that allow repeat the experiment.
Here my minor comments
Response: Thank you very much for your helpful comments. We have improved the ms according to your suggestion. Please see the ms revision and the responses below.
Line 108-109: in how many nest you recorded the alarm calls
Response: Thank you for your advice. We recorded 12 nests of vinous throated parrotbill (VP) and 11 nests of oriental reed warbler (ORW). Please see line 160.
Line 109: Do you use the internal microphone? was it a mono or stereo recording? File format?
Response: Thank you for your question. We used an internal microphone, which was stereo recording, and the file format is wav. Please see line 162.
Table 1: n is the number of nests or the number of calls analyzed?
Response: Thank you for your advice. N is the number of nests.
Line 123-124: this sentence is not clear. You mixed in a single playback stimulus call from 3 adults? If this is the case, it is a very uncommon stimulus type, because you are producing stimulus from 3 different individuals, which is not common in nature. So, you need to justify this uncommon stimulus design.
Response: Thank you for your suggestion. To avoid pseudo-replication, we mixed the calls of three adults to produce a single playback stimulus call. If we had used a single playback from one adult, pseudo-replication might have occurred in biostatistics. Indeed, the aim of our study was to test the function of alarm calls, so we ignored the difference in calls between individuals.
Line 127: which filter type you used
Response: Thank you for your advice. We did not filter the calls, but cut off the noise below 0.2 kHz to avoid the noise interference of the wild.
Line 135: which is the SPL value? You need to report the value in dB.
Response: Thank you for your advice. We express the SPL values that we measured in wild in lines 187-188. And SPL values of playback stimulus approach the condition of wild when we conduct the playback experiment.
Lines 137-138: 54.43 and 74.2 are? Units
Response: Thank you for your advice. We have mentioned the units in line 188.
Line 154: a single stimulus has the three stimuli in sequence? Explain this very well
Response: First, a natural begging without sound playback was recorded, and then one of three playback sounds was played at random, five minutes apart. Please see lines 205–209.
Lines 154-156: these sentences need to be explained in more detail because it is not clear what you did.
Response: Thank you for your suggestion. We have added the purpose and spectrograms of alarm calls, which was used in the playback. Please see lines 207-209 and spectrograms.
Reviewer 2 Report
See attached file

Author Response
Reviewer 2:
Overall appraisal:
The main value of this study lies in the experimental demonstration of a presumably
adaptive anti-predation response (reduced conspicuous begging behaviour) by chicks
induced by heterospecific mobbing alarm calls normally given by adults. Although there has been a considerable body of published research in this general area, this investigation makes a valuable and interesting contribution to knowledge of anti-predator responses of avian nestlings.
Response: Thank you very much for your helpful comments. We have improved the ms according to your suggestion. Please see the ms revision and the responses below.
Detailed appraisal:
L 16. Predation is generally the main cause ………
Response: Thank you for your advice. We have corrected this sentence. Please see line 59.
L 37-38. Say briefly how these differ, in particular how do distress calls work?
Response: Thank you for your suggestion. We have added the information of these three calls. Please see introduction.
L 40-41. Some redundancy here – streamline
Response: Thank you for your advice. We have corrected this sentence. Please see lines 87-89.
L 50. Sciuru or Sciurus?
Response: Thank you for your advice. We have corrected this word. Please see line 97.
L 53. Evolutionary forces? You mean natural selection presumably?
Response: Thank you for your question. Here, evolutionary forces come from theevents of predation. These prey, which have similar hunters, may evolve the ability to understand heterospecific calls.
L 71-74. Needs a bit more explanation
Response: Thank you for your advice. We have added the relevant information. Please see lines 120-122.
L 79. Delete second “to”
Response: Thank you for your suggestion. We have deleted this word. Please see line 128.
L 104-105. What is a mosaic pattern in this context?
Response: Thank you for your question. Mosaic pattern means the nests of these two species are distributed in the same habitat and are not non-overlapping.
L 108. Would be useful to have a figure showing spectrograms of calls.
Response: Thank you for your advice. We have added a Figure S1 to the supplementary data.
L 110. Do you know that MAC given to a researcher is the same as that given to nonhuman predators? Do these species actually mob humans near nests?
Response: Thank you for your question. We believe the approach of a researcher or nonhuman predators belongs to the risk of predation. Both species can mob humans near nests according our observations.
L 117. Define Delta frequency
Response: Thank you for your suggestion. Delta frequency is the frequency between two audio frames.
L 126. What does chip the sound mean?
Response: Thank you for your question. Chipping the sound means that we used Raven pro to cut out the calls to remove noise below 0.2 kHz.
L 152-156. Can you make this clearer? Maybe a timeline diagram would help?
Response: Thank you for your suggestion. We have provided more details in lines 206-209.
Figs 1-3. I do not find these figures easy to follow; I suggest that you simplify them
Response: Thank you for your suggestion. The figures aim to express the comparison between controland three trials in each species. We did not conduct comparison between species. Thank you again!
L 231. Surely an innate response is more likely than a learned one in such young birds facing such a strong threat to survival?
Response: Thank you for your question. Compared with a learned behavior, we speculate that an innate response may play an important role in reducing the risk of predation. This is because the events of predation are quick and costly, so an innate response can enhance the overall fitness maximum.